

# Virgin and UV-weathered polyamide microplastics posed no effect on the survival and reproduction of *Daphnia magna*

Alla Khosrovyan[1] and Anne Kahru[1,2]

[1] Laboratory of Environmental Toxicology, National Institute of Chemical Physics and Biophysics, Tallinn, Estonia
[2] Estonian Academy of Sciences, Tallinn, Estonia

## ABSTRACT

Although evidence suggests that microplastic (MP) particles pose a risk to organisms, the effects of virgin and weathered MP should be evaluated separately as their effects may be different. In this work, we provide new information on the toxic potential of virgin and UV-weathered polyamide, one of the commonly used plastics worldwide. Polyamide MP particles were subjected to UV-weathering in wet conditions over 26 days in a customized irradiation chamber equipped with UV-C light tubes (15 W each, max. wavelength 254 nm). The toxicity of virgin and UV-weathered polyamide MP ($<$ 180 $\mu$m in one dimension, 100 and 300 mg L$^{-1}$) was evaluated by studying *Daphnia magna* reproduction in natural lake water spiked with MP, following the 21-day OECD 211 test guideline. In parallel, a nonionic surfactant Tween 20 (7 mg L$^{-1}$) was added to the test medium to improve the suspendability of the MP. The results of the tests showed no adverse effects of either virgin or UV-weathered polyamide MP on the reproduction of *D. magna*. In addition, presence of Tween 20 in the test medium had no effects on the test results. These results bring a new perspective on the potential long-term impact of polyamide particles on aquatic organisms, especially considering that the polyamide has received marginal attention in the ecotoxicological research. However, standard test endpoints (survival and reproduction) may still miss long-term adverse effects of insoluble *e.g.*, plastic particles and additional studies may be necessary.

## INTRODUCTION

Recent ecotoxicological studies on the effects of microplastics (MP) (plastic particles <five mm in one dimension) in aquatic systems have included more freshwater organisms whereas previously marine systems/organisms were the primary focus (*Castaneda et al., 2014*; *Hurley, Woodward & Rothwell, 2018*; *Khosrovyan & Kahru, 2020*; *Khosrovyan & Kahru, 2021*; *Nel, Dalu & Wasserman, 2018*; *Silva et al., 2021*; *Wang et al., 2020*). Indeed, the increased impact of plastic pollution in freshwater systems has been reported in

Corresponding author
Alla Khosrovyan, alkhosrov@yahoo.com, alla.khosrovyan@kbfi.ee

rivers and lakes worldwide (*Baztan et al., 2014*; *Klein, Worch & Knepper, 2015*). However, many knowledge gaps still exist. For example, not all widely-used plastic types have been investigated for their potential adverse effects: the available ecotoxicological data is biased towards particular MP types *e.g.*, polyethylene (PE) ∼7%, polystyrene (PS) ∼2%, polypropylene (PP) and polyvinyl chloride (PVC) ∼1% and polyamide (PA) ∼0.5% (data from the Web-of-Knowledge topic search performed by the authors using *plastic* (or a particular type of it), test*, toxic*). In addition, research on the potential toxicity of UV-weathered plastics and their leachates is also inadequate, even though photo-oxidation by UV light is the primary cause of plastic degradation. A recent search on the Web-of-Knowledge indicated that fewer than 2% of the studies on plastics toxicity considered UV-weathered plastics. Photooxidation of the plastics surface following UV irradiation results in the polymer breakdown and embrittlement causing further fragmentation of the plastics by lower energy levels (*Huffer, Weniger & Hofmann, 2018*) and surface functionalization fostering absorption of not only hydrophobic but also hydrophilic organic pollutants (*Liu et al., 2019*). UV-weathered PVC beads (50–100 µm) were more toxic to the freshwater algae *Chlamydomonas reinhardtii* than virgin PVC beads, due to the formation of carbonyl groups on the surface of the beads (*Wang et al., 2020*). Leachate from UV-irradiated plastics of various types released plasticisers at concentrations comparable to those measured in the riverine water in the United States and caused significant vitellogenin induction in Japanese medaka *Oryzias latipes* larvae compared to virgin plastic leachate (*Coffin et al., 2018*). *Khosrovyan & Kahru (2021)* reported significant reduction in the emergence of midges *Chironomus riparius* exposed to UV-weathered polyamide particles compared to virgin particles.

Further, the impact of ingested MP particles is not easily predictable. For example, despite a greater number of ingested polyamide MPs at higher concentration (1000 mg kg$^{-1}$ of dry sediment) compared to lower concentration (100 mg kg$^{-1}$ of dry sediment), the number of emerged *C. riparius* adults in both treatments was similar, presumably due to the successful excretion of the particles from their gastrointestinal tract (*Khosrovyan, Gabrielyan & Kahru, 2020*). Localized leaching of monomers and additives in the gastro-intestinal system of organisms may also be responsible for the chemical toxicity of ingested particles (*Wright & Kelly, 2017*). Several studies have demonstrated that the leachate obtained after 24 h of contact with virgin polyethylene pellets or different plastic products with seawater was toxic to aquatic organisms such as barnacle (*Amphibalanus amphitrite*) nauplii (*Li et al., 2016*), brown mussel (*Perna perna*) larvae (*Gandara e Silva et al., 2016*) and sea urchin (*Lytechinus variegatus*) larvae (*Nobre et al., 2015*).

In this study, the potential adverse effects of virgin and UV-weathered polyamide MP of irregular shape and different sizes were evaluated using the OECD 211 *Daphnia magna* reproduction test. *D. magna* is one of the key freshwater organisms filtering water column. Our working hypotheses were as follows: (i) virgin particles are harmful to daphnids over the long-term exposure, (ii) UV-weathering promotes degradation of the particles and/or leaching chemicals from them, and (iii) UV-weathering of the particles augments the hazard of the particles to aquatic biota. A comparison with the effects of virgin and UV-weathered polyamide particles on another key freshwater/sediment organism—*C.*

*riparius*—evaluated using the OECD 233 test guideline (*Khosrovyan & Kahru, 2021*) is provided.

## MATERIAL AND METHODS

### Virgin and UV-weathered microplastics

Virgin polyamide particles (PA-MP) were from the batch described in our previous work (*Khosrovyan & Kahru, 2020*; *Khosrovyan & Kahru, 2021*). Specifically, they were bought from a commercial supplier (Abifor AG, Baden-Württemberg, Germany). They had irregular shape and size range from 0 to 180 µm with 88% of particles being <160 µm. Particles were fluorescent which made their observation in the animal gut under fluorescent microscope possible without any modification. Information on fluorescent dye and additives included in the particles was not provided by the supplier.

UV-weathered particles (UV-PA-MP) were obtained by a process of artificially aging/weathering of virgin particles (PA-MP) in a customized temperature-controlled UV-C light irradiation chamber for 26 days. Measures to avoid dusting of the particles inside the chamber were applied: chamber door was kept shut and the chamber was placed in a closed room. Temperature in the chamber during the exposure ranged from 33−35 °C. Virgin PA-MPs were placed in a glass bowl filled with dechlorinated tap water and exposed to five new UV-C light tubes (15 W each, max. wavelength 254 nm), mixing the particles in the bowl from time to time to ensure equal water and UV light exposure conditions and thus, uniform aging. This wavelength was selected based on higher electron volt energy compared to UV-A band (http://www.uvresources.com/) and previous reports (*Huffer, Weniger & Hofmann, 2018*; *Liu et al., 2019*; *Khosrovyan & Kahru, 2021*). Before use of the aged particles in the experiments the particles were air-dried.

For detecting changes on particle surfaces after UV-C irradiation, surface scans of 20 particles of each type (PA-MP or UV-PA-MP) were obtained using a Zeiss Confocal laser scanning microscope LSM 800 (objective Plan-Apochromat 20x/0.8 M27, laser wavelength 404 nm, detection wavelength 400–531 nm) and images before and after UV irradiation visually inspected under the microscope and compared.

For quantification of the increase of the share of small-sized particles as the result of UV-weathering, particles of each type (virgin and UV-weathered) were placed on the slides of an imaging microscope (Nikon SMZ1270) and those ranging from 10–40 µm size were counted by random application of 1 × 1 mm frames to each slide (10 frames per type, ×3 magnification).

### Test organism and exposure scenarios

The water flea *D. magna* was obtained from an in-house facility originated from ephippia purchased from Microbiotests, Inc. (Gent, Belgium), batch number: DM020421. The *D. magna* were cultured in glass vessels at 20 ± 1 °C (16:8 h light:dark period, 1000 lux) in the medium used in the assays. Daphnids were fed with algae *Raphidocelis subcapitata* suspension *ad libitum*. Culturing conditions followed the OECD 211 *D. magna* reproduction test guidelines (*OECD, 2012*). On reaching maturation, several parental

daphnids were separated into the mother stock facility for producing offspring to be used in the experiments.

Two exposure scenarios in two different media were used: natural water medium (hereafter, water-only) and the same medium spiked with a surfactant (hereafter, surfactant-including). Natural water from Lake Ülemiste (Estonia) was provided by the water company "Tallinna Vesi". The water was immediately passed *via* 0.45 μm Millipore filter and stored for a few days in the dark at 4 °C until used in the bioassays. The physico-chemical characterization of Lake Ülemiste water is presented in Table S1. As a non-ionic surfactant, Tween 20 (Sigma-Aldrich, St Louis, MO, USA) was used.

## Experiments

Toxicity tests were conducted according to adapted OECD 211 *D. magna* reproduction test guidelines (*OECD, 2012*): semi-static test with water renewal every third day at a temperature of $20 \pm 0.5$ °C was performed in an Algaetron incubator (Photon Systems Instruments, Drásov, Czechia), photoperiod 16:8 h light:dark, light intensity 1000 lux, pH within the range 6–9. Ten newborn daphnids (<24 h old, not the first brood) were placed individually into ten 50 mL glass vials (one daphnia per vial, 10 replicates for the controls and treatments). Each vial contained 50 mL of exposure medium: Ülemiste water or Ülemiste water spiked with a surfactant. As a surfactant, Tween 20 (Sigma-Aldrich, St Louis, MO, USA) was used at 7 mg $L^{-1}$ (0.0007% v/v). This concentration was not toxic to daphnids and was selected based on the results of a duplicated 48 h acute immobilization test according to OECD 202 guidelines (*OECD, 2004*) which revealed mortality of 1/10 daphnids in 1/10 replicates at 130 mg Tween $L^{-1}$ (in the Ülemiste water). Mortality after 96 h was 1/10 and 4/10 (in the test and its repetition, correspondingly).

The medium of each experimental vial was spiked with a nominal PA-MP or UV-PA-MP concentration of 100 or 300 mg particles $L^{-1}$. In the water-only scenario, to each vial PA-MP or UV-PA-MP particles were added at both concentrations (100 and 300 mg $L^{-1}$), while in the surfactant-including scenario, only UV-PA-MP at 300 mg $L^{-1}$. This procedure was repeated at each medium renewal (every third day). Because of frequent medium renewals no additional measurement of exposure concentrations and comparison with the nominal concentrations was performed. As a negative control, filtered natural water was used. An additional surfactant control was run for the surfactant-including test at 7 mg Tween 20 $L^{-1}$.

During the first days each daphnid was fed 55 μl of the same algal suspension as in the culture (*R. subcapitata*) and on reaching the adult age this volume was increased to 98 μL (such a volume corresponded to the daily carbon requirement of 0.1 mg C/*Daphnia* based on optical density measurement conducted daily in our laboratory). Following the guidelines, the exposure medium was regularly renewed every third day during the 21 days of exposure (six times in total). At renewals, daphnids were carefully transferred to new vials filled with aerated fresh medium and/or polyamide particles. The test endpoints were the number of surviving parents at the end of the experiment and the number of offspring per live parent.

Daily, animals were monitored for any unusual swimming behavior, parental mortality was recorded, dead animals removed and offspring counted. pH of the medium when renewed was randomly verified during the experiments.

The validity of the tests was verified in accordance with the OECD 211 guidelines for the controls: the mortality of parental daphnids should not exceed 20% and the mean number of living offspring per surviving parent should be $\geq$ 60 at the end of the test.

## Statistical analysis

Significant differences in the reproduction between the control and treated animals (within experiment) and between similar treatments from two independent experiments (between experiments) were analyzed by means of one-way ANOVA ($p < 0.05$) run on R (version 4.1.3 (2022-03-10)) using *rstatix* library. Mean and standard deviation were calculated with MS Excel. 48-h and 96-h LC50s of Tween 20 dispersed in natural water (Lake Ülemiste) for *D. magna* were calculated with MS Excel macro Regtox (*Vindimian, 2016*), commonly used in ecotoxicology for estimating the dose–response relationship. The concentration-effect curves were derived using five nominal exposure concentrations (7, 15, 30, 60, 130 mg Tween 20 $L^{-1}$) *vs* the mortality of *Daphnia* and LC50s were calculated based on the log-normal model.

The number of particles of each plastic type (virgin and UV-weathered, 10–40 μm) counted with the help of an imaging microscope was analyzed for mean difference by applying the independent samples *T*-test run on R ($p < 0.05$). Normality and homogeneity of variance was verified prior to the *T*-test.

## RESULTS

### Post-aging changes in the MP particles

During artificial aging in the UV-C light irradiation chamber, the color of the virgin particles changed from white to light yellowish/brownish (Fig. 1). The buoyancy of particles was also affected: initially fluffy particles floating on the water surface sank to the bottom by the 10th day of irradiation. The particles also became susceptible to breaking into small pieces when smashed manually by a stone pestle: the size range of UV-weathered particles included a significantly larger fraction of smaller particles (10–40 μm) in contrast to virgin PA-MP particles (means: 89 and 33.1, respectively; t = −5.4, *p*-value = 3.636e−05).

However, a microscopic examination of in total, 40 particles (20 per type) for surface changes likely caused by UV-C weathering did not reveal any distinguishable changes. For example, surfaces of all particles (before and after weathering) had similar wrinkles, cracks and deepenings and they hosted bunches of smaller particles. Microscope images of the surfaces of four particles of each type (PA-MP or UV-PA-MP) are shown in Fig. S1.

### Life–cycle effects

The effects on *D. magna* caused by polyamide particle ingestion were evaluated using the standard test (OECD 211) endpoints: survival and reproduction. Ingestion of particles was verified *via* a separate 2-day exposure investigation with *D. magna* as shown in Fig. 2 depicting the particles in the gut of the animal under Zeiss LSM 800 laser scanning

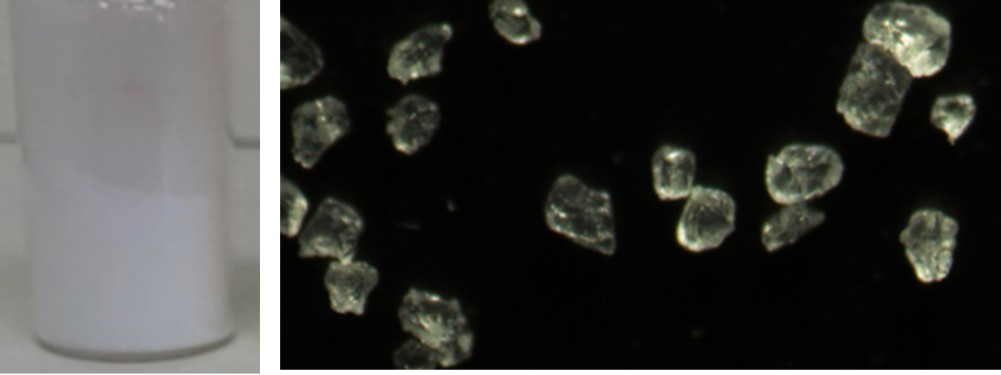

Virgin polyamide particles

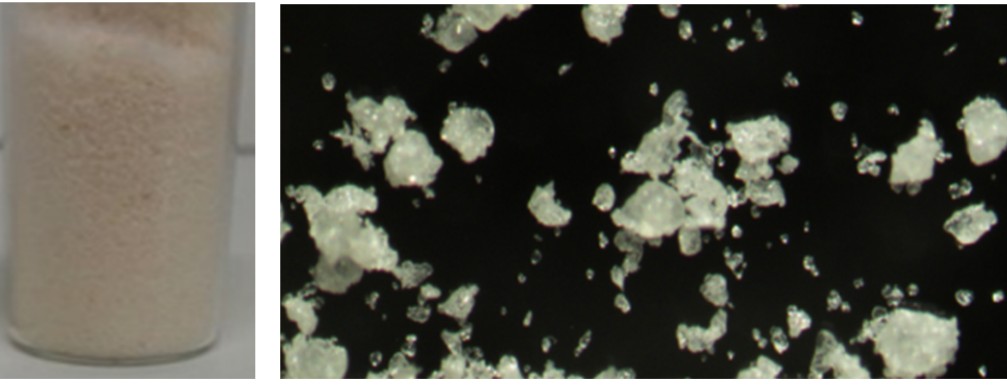

UV-weathered polyamide particles

**Figure 1** **Virgin polyamide particles and the change of colour and increase in the fraction of smaller particles after UV-C weathering.**

microscope LSM 800. The particles were clearly visible in the gut due to their fluorescence (blue particles in Fig. 2). Quantification of ingested, accumulated or eliminated particles from the daphnids was out of scope of this work.

However, despite the evident ingestion of the particles, no significant differences in the number of offspring produced by daphnids were seen between the treatments and their corresponding controls in both experiments (water-only and surfactant-including). No mortality was observed in the water-only experiment, except for three replicates (10% in each). In the surfactant-including experiment, survival of parental daphnids was 100%. Also, no significant differences were observed between the experiments *e.g.*, between PA-MP treatments from the two experiments (duplicated experiment). The details of the test of difference are presented in Table S2. As no difference was detected between experiments, the data for the corresponding treatments and controls were pooled and averaged, Fig. 3.

The initial pH of the natural water was 8.32 and after addition of the surfactant 8.38. Random measurements of pH of the medium at renewals demonstrated negligible variations (8.4 ± 0.03 in the water-only and 8.3 ± 0.1 in the surfactant-mediated

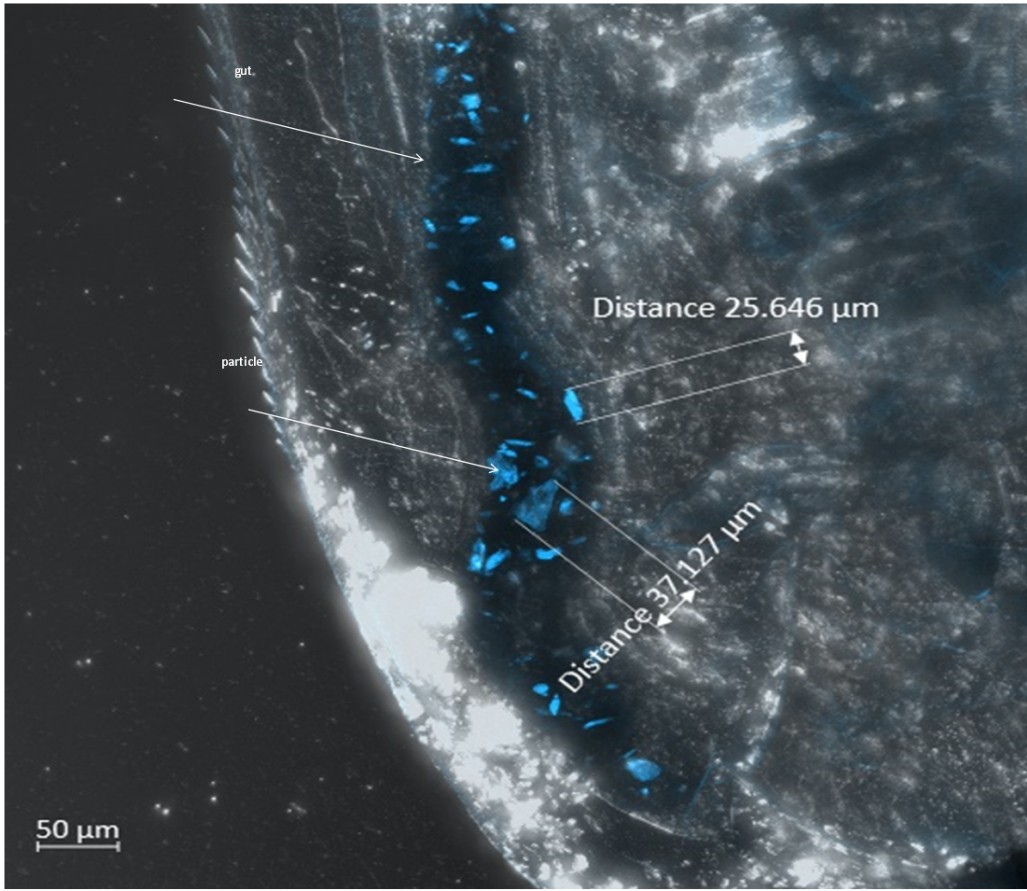

**Figure 2  UV-weathered fluorescent polyamide particles in the gut of *D. magna* (image from the Zeiss confocal laser scanning microscope LSM 800, laser wavelength 405 nm, detection wavelength 400–531 nm).** Arrows point to the gut (up) and polyamide particle (bottom). Particle size is indicated by distance measured between two edges of the particle.

experiments), which conformed to the test guidelines: pH within the range 6–9 and no variation by more than 1.5 units in any test (*OECD, 2012*).

## DISCUSSION

In this study, newborn daphnids were exposed for 21 days to virgin and UV-weathered PA-MP particles at two concentrations (100 and 300 mg L$^{-1}$) under two exposure scenarios (with and without surfactant), in accordance with the OECD 211 *D. magna* reproduction test guideline. The conditions for UV weathering of the PA-MP particles (open air and in water environment) were chosen to resemble environmentally relevant ones since discarded plastics often end up in surface water where it undergoes exposure to UV radiation, wind and wave action and abrasion (*Wright & Kelly, 2017*). UV weathering is accelerated by exposure to the air (*Gijsman, Meijers & Vitarelli, 1999*) and in humid conditions (*Singh & Sharma, 2008*). Also, at any given temperature and humidity, the rate of degradation increases with increasing UV flux (*Singh & Sharma, 2008*). UV weathering

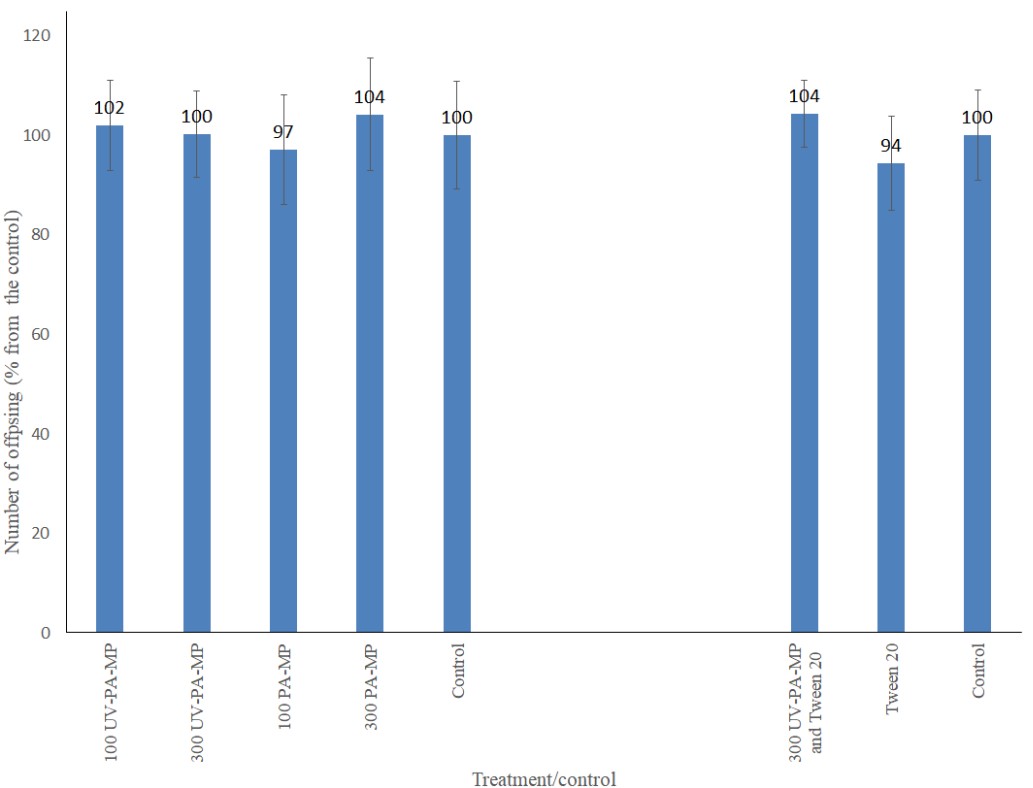

**Figure 3** **Mean number of offspring per surviving parent** *Daphnia* **(% from the control) at the end of the 21-day exposure to two types of polyamide: virgin (PA-MP) and UV-weathered (UV-PA-MP).** Number above each data column: the actual percentage. Bars: standard deviation. Treatments: virgin and UV-weathered polyamide at concentrations of 100 and 300 mg L$^{-1}$ with and without surfactant (7 mg Tween 20 L$^{-1}$). Data are averaged from the duplicated experiments.

occurs due to impurities embedded in the polymer matrix that can absorb photons, which are excited and cleave the polymer chain (*Singh & Sharma, 2008*). The presence of oxygen initiates photooxidative degradation: a radical-based autoxidative process during which free radicals (formed due to a photolysis reaction of an impurity) propagate by thermal reaction (stimulated by the ease of hydrogen abstraction from the carbon-hydrogen bond of the polymer). At the final stage, peroxy radicals cause further reaction leading to the formation of dialkylperoxides, alcohols and ketones (*Gijsman, Meijers & Vitarelli, 1999*). Hence, the formation of different substances on the surface of the UV-PA-MP particles used in this study was expected to cause some toxicity to the daphnids upon particle ingestion. In addition, photooxidation of plastic particles makes them brittle and susceptible to degradation into smaller fragments (*Jahnke et al., 2017*; *Wright & Kelly, 2017*). Hence, an increase in the fraction of smaller particles in the pool of particles of different sizes due to UV weathering (Fig. 1) was expected to increase their bioavailability and contribute to the toxicity to the daphnids.

The particle concentrations chosen for this study were in the range used in testing of the responses of the *D. magna* to MP exposure by previous researchers, *e.g.*, 100 mg

$L^{-1}$ PE terephthalate textile fibers (PET) (*Jemec et al., 2016*) or PE microbeads at 100 mg $L^{-1}$ (*Canniff & Hoang, 2018*) or at 0.0001–10 g $L^{-1}$ (*Frydkjær, Iversen & Roslev, 2017*) or 100 mg $L^{-1}$ polystyrene (PS) nanoplastics explored in nine bioassays including *D. magna* (*Heinlaan et al., 2020*). The consensus is that the current environmental concentrations are not harmful *e.g.*, to microalgae (*Prata et al., 2019*) or as a vector for hydrophobic pollutants (*Frydkjær, Iversen & Roslev, 2017*). However, the actual concentrations of MP in aquatic environments remain largely unknown. They can vary *e.g.*, from 257–1215 pieces $m^{-3}$ along the south-eastern coastline of South Africa (*Nel & Froneman, 2015*) to 30 g $L^{-1}$ on the high tide line in Fuerteventura beaches in the Canary Islands in Spain (*Baztan et al., 2014*). Normally, the MPs sampled from aquatic environments contain particles of different types, shapes and sizes (usually disregarding low micrometer ranges) and the determination of environmental concentrations depends among others on the geomorphological profile of sampling sites, sampling tool, mesh size, duration and depth of sampling, extraction and identification methods. For example, geomorphological conditions (such as rocky shorelines with little sand deposition) and high energy waves can affect the deposition of MP particles on beaches (*Baztan et al., 2014*) and hence, their availability for sampling. In addition, MP concentrations tend to increase because of the wide applications of microbeads in various spheres and the degradation of the existing plastics. Hence, the effects of the supposedly higher concentrations tested in laboratory conditions worldwide help proactively estimate the future impacts of the plastics on the biota.

Nevertheless, the results obtained for the water-only scenario, when daphnids were continuously exposed to virgin and UV-weathered PA-MP particles through their life cycle (21 days), showed no effect on their survival or number of offspring produced (Fig. 3), even though the UV-weathered PA-MP particles were more abundant at 10–20 μm (compared to the virgin ones) and small particles can be more easily ingested by non-selective filter-feeders (*e.g.*, *Desforges, Galbraith & Ross, 2015*; *Rehse, Kloas & Zarf, 2016*).

The addition of Tween 20 to the medium (*i.e.*, surfactant-including scenario) was used to ensure that the plastic particles are well mixed within the medium and will be readily bioavailable to *D. magna* throughout the experiments. While *D. magna* can feed from the bottom (authors observations, also *Imhof et al., 2013*; *Imhof et al., 2017*) and can swim into the particle layer on the water surface (authors observations, also *Rehse, Kloas & Zarf, 2016*), the addition of the surfactant was expected to improve the contact of particles with water and ensure their homogeneous mixing in the water column and thus availability to the filter-feeding *D. magna*. *Frydkjær, Iversen & Roslev (2017)* also used Tween 20 at 0.001 vol% to facilitate mixing of PE and water in the ingestion study with *D. magna*. Besides, the scenario with the surfactant possesses high environmental realism as nonionic surfactants have found wide application in various spheres *e.g.*, as detergents, emulsifiers, cosmetic and pharmaceutical components (*Ferreira, Bernin & Topgaard, 2013*). They can be found in the environment and can modulate the bioavailability of MP (*e.g.*, by improving the contact among MP and animals and dispersing MP in the water column *Renzi, Grazioli & Blašković, 2019*), in addition to posing a direct toxicity risk to aquatic animals (*Garcia, Campos & Ribosa, 2007*; *Ivanković & Hrenović, 2010*). For example, according to *Renzi, Grazioli &*

*Blaškovic (2019)*, the addition of a surfactant (Triton X-100) equal to 0.001% v/v to various MP dispersions (PE, PP, PVC, PVC/PE) increased mortality and immobilization of early stages of *D. magna* under both fasting and feeding conditions during 96 h of exposure (MP range size 10–100 $\mu$m and concentration 50 mg L$^{-1}$).

The results of 48 and 96-h LC50 values of Tween 20 (>130 mg L$^-$1) and of the 21-day life cycle experiment at 7 mg Tween 20 L$^{-1}$ (alone and in combination with UV-PA-MP) showed no effect on the survival or number of offspring produced by *D. magna* (Fig. 3). Indeed, according to the hazard classification of the European Union legislation 2008 to the aquatic environment (categories I–III), Tween 20 cannot be categorized even as category III (LC50 >10 to $\leq$ 100 mg L$^{-1}$). However, effects of surfactants depend on species and surfactant type and the effect concentrations of nonionic surfactants ranged from 0.3 to 400 mg L$^{-1}$ (*Evsyunina et al., 2016*; *Jahan, Balzer & Mosto, 2008*; *Liwarska-Bizukojc et al., 2005*; *Pettersson, Adamsson & Dave, 2000*).

Effects caused by MP particles ingestion may not be restricted to the particles themselves. Leachate of toxic additives embedded in the plastic polymer matrix to impart desired qualities (*Bejgarn et al., 2015*; *Hahladakis et al., 2018*) may also pose a risk to the biota. Such toxic additives may be loosely bound to the structure and thus can easily leach out from the polymer. According to the data, the leachate obtained from different plastic products cut into 10x10 cm pieces during 24 h at 28 °C (PE, PET, PS, PP, PVC, polycarbonate - PC) significantly lowered survival and settlement of barnacle *Amphibalanus amphitrite* larvae at concentration 0.10 and 0.50 m$^2$L$^{-1}$ (*Li et al., 2016*). Leachates from 9/32 plastic products prepared on a rotator for 24 h at ~20 °C had 48-h *D. magna* immobilization EC50s ranging from 5 to 80 g of plastic material L$^{-1}$ (for PC, PVC, polyurethane) (*Lithner et al., 2009*). Furthermore, leachate obtained from PVC and epoxy products (2 × 2 cm pieces) over 3 days at 50 °C had 48-h *D. magna* immobilization EC50s in the range 2–235 g of plastic material L$^{-1}$ (*Lithner, Nordensvan & Dave, 2012*). Although the PA-MP concentrations used in our experiments were much lower than the above-mentioned estimates of toxicity (24-h and 48-h EC50s), the time for leachate formation was longer in our case (72 h between medium renewals) and some toxicity of the leachate was expected at least for the earlier life stages of the daphnids. However, insignificant differences in the number of offspring between the controls and treatments in both (water-only and surfactant-including) experiments suggest no toxic potential of the leachates. Apparently, particles did not contain leachable toxic additives or at the selected concentrations 72 h (in between water renewals) was not sufficient for the formation of toxic leachate.

On the other hand, abundant studies indicate low MP effects on *D. magna* where comparable particle size range, type, concentrations, and exposure conditions were tested (*e.g.*, *Aljaibachi & Callaghan, 2018*; *Jemec et al., 2016*). *D. magna* was shown to ingest 63–75 $\mu$m PE beads (25, 50, and 100 mg L$^{-1}$) and although their gut was filled with them, no significant effect on survival and reproduction was seen (*Canniff & Hoang, 2018*).

The particles used in the study were irregularly shaped and thus their egestion from the gut of daphnids could be complicated, compared to regular particles, thus supposedly exerting additional stress on them. *Frydkjær, Iversen & Roslev (2017)* showed that the full egestion of irregular PE particles (10–75 $\mu$m) by *D. magna* was significantly lower

than that of spherical PE beads (10–106 µm) after 24 h of exposure at concentrations 10, 100, 1000 and 5000 mg L$^{-1}$. Although MP particle morphology can play a role in particle uptake and evacuation from the organism, our results showed no effect of the used irregular shape (virgin and UV-altered) polyamide MP on the reproduction of the *D. magna*. This largely agrees with reported data of no acute toxicity to *D. magna* of irregular shape MPs at <100 µm size at 100 mg L$^{-1}$ (*Kokalj, Kunej & Skalar, 2018*) or no effect on reproduction or malformation of juveniles caused by irregular particle mixtures 22–72 µm size at concentrations of 1% of the food particles although with some inconsistent alterations in stress response and gene expression (*Imhof et al., 2017*). A contrasting variety of experimental outcomes with *D. magna* on MP ingestion (from increased mortality to no effect) was also pointed out by *Bosker et al. (2019)*.

When comparing the responses of two key aquatic species (*D. magna* and *C. riparius*) to virgin and UV-weathered polyamide exposure, one can see that the standard endpoint in the midge *C. riparius* (emergence of winged adult midges) was also not affected when the midges were exposed to 100 or even 1000 mg kg$^{-1}$ of virgin PA-MP during life cycle experiments (*Khosrovyan & Kahru, 2020*; *Khosrovyan & Kahru, 2021*). In a multi-generation experiment, emergence was reduced after the first generation but recovered during the second and third generations (*Khosrovyan & Kahru, 2021*). However, in contrast to *D. magna*'s response to exposure to UV-PA-MP (at 300 mg kg$^{-1}$), significantly lower emergence, compared to the control, was seen in *C. riparius* exposed to UV-PA-MP (1,000 mg kg$^{-1}$) throughout its life cycle. Ingestion of the polyamide particles was verified for both organisms (*Khosrovyan, Gabrielyan & Kahru, 2020*; Fig. 2). That no toxicity of virgin polyamide particles at the *Daphnia*'s organism level was observed in this study may suggest that they were efficiently removed from the body after being ingested and/or did not contain leachable toxic additives. However, it remains unclear what can account for the adverse effect of UV-weathered particles on *C. riparius* and no adverse impact of the same particles on *D. magna*.

Responses to virgin or weathered MP may be species-specific, even if the species are standard test organisms. Different dietary habits (*e.g.*, benthic gathering *vs* filter-feeding) influence the amount and possibility of the uptake of solid particles and further going through the digestive tract of these items. Hence, overgeneralization of response variables may occur if the species-specificity to response or types of exposure (external or ingested) is disregarded (*Suckling, 2021*). However, more sensitive endpoints in microplastic exposure studies must be considered to improve the performance of standard tests.

## CONCLUSIONS

Exposure to virgin and UV-weathered polyamide particles at 100 and 300 mg L$^{-1}$ with and without Tween 20 surfactant did not affect the survival or number of offspring produced by the *D. magna*. The no adverse effect of particle ingestion on *D. magna*'s survival and reproduction may suggest that virgin plastic particles lacking toxic additives do not pose a toxicity risk to them. This can also be supported by no long-term adverse effects of virgin particles on *C. riparius* in a multi-generation study.

However, the toxicity of pure MP particles should still not be overlooked because in the environment they adsorb contaminants and/or undergo degradation and this entails corresponding consequences for the biota. Although the impact on the biota may depend on the species and types of exposure, the existing standard tests may not perform well in the case of insoluble particulates such as MP particles and new endpoints could be suggested for detecting MP effects.

Low concentrations of Tween 20 (*e.g.*, 7 mg L$^{-1}$) were nontoxic but sufficient for altering surface tension of polyamide particles (and likely, other polymer) particles without causing confounding effects on the crustacea and it can be recommended for use in plastics toxicity studies.

## ACKNOWLEDGEMENTS

The authors thank Anna Schemmer, MSc student from University Koblenz–Landau (Germany), for help during her Erasmus+ internship at the Laboratory of Environmental Toxicology, National Institute of Chemical Physics and Biophysics, Tallinn, Estonia.

### Funding

The project was funded by the Estonian Research Council grants (Mobilitas Pluss MOBJD509, TT13 and PRG749) and European Regional Development Fund grants (NAMUR+ 2014-2020.4.01.16-0123 and TK134). The funders had no role in study design, data collection and analysis, decision to publish, or preparation of the manuscript.

### Grant Disclosures

The following grant information was disclosed by the authors:
The Estonian Research Council grants: Mobilitas Pluss MOBJD509, TT13 and PRG749.
European Regional Development Fund: NAMUR+ 2014-2020.4.01.16-0123 and TK134.

### Competing Interests

The authors declare there are no competing interests.

### Author Contributions

- Alla Khosrovyan conceived and designed the experiments, performed the experiments, analyzed the data, prepared figures and/or tables, authored or reviewed drafts of the article, and approved the final draft.
- Anne Kahru analyzed the data, authored or reviewed drafts of the article, and approved the final draft.

### Data Availability

The raw experimental data are available in the Supplemental Files.

## Supplemental Information

Supplemental information for this article can be found online at http://dx.doi.org/10.7717/peerj.13533#supplemental-information.

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
