# Peer review of "Virgin and UV-weathered polyamide microplastics posed no effect on the survival and reproduction of Daphnia magna"

_PeerJ, doi:10.7717/peerj.13533_

## Round 0.1 · original submission · Minor Revisions

Two Reviewers have provided their opinions of the manuscript and both were in favor of its publication. Please address the Reviewers' suggestions before publication. In addition to Reviewers' comments, please address the following:

The title of the manuscript is in a question format that can be answered either “yes” or “no”. Since the issue is likely more complicated and can be affected by several factors, thus, likely does not have a simple “yes” or “no” answer, please rephrase the title to avoid this issue.

In the abstract, information about the preparation process of UV-weathered (or UV-degraded) polyamide microplastics (MP) could be added, as well as how was it confirmed that MPs were UV-degraded. Were the virgin and degraded MPs confirmed to be clearly different in their properties?

Line 74, Daphnia magna should be D. magna.

Line 94, add the brand and manufacturer of the irradiation chamber.

Lines 196-172, this method should be included in the section “Virgin and UV-weathered microplastics”, not in “Statistical analysis”. In addition, please explain what method and criteria were used for the comparison of images before and after UV irradiation. Was quantitative analysis conducted and how? What were the criteria to decide if MPs were UV-degraded or not?

Lines 181-182, “…the size range of UV-weathered particles included a larger fraction of smaller particles (10-20 μm) in contrast to virgin PA-MP particles.” How was this quantitative analysis conducted? Please add the method description in the Methods section.

Reviewer 1 ·

Basic reporting

The manuscript is clearly written with supporting contextualizing references. The results are relevant despite the absence of effects. Negative results are extremely important to assess MPs toxicity.

Experimental design

The research question is well defined and the methods are adequate by following guidelines. All technical aspects are correct and detailed.

Validity of the findings

The findings point to no toxicity of PA MPs. Negative results are also very important for MPs toxicity and the authors present sufficient evidence from the literature supporting their results.

Additional comments

In my opinion, the manuscript is well written and has the potential to be published in this journal.

·

Basic reporting

This manuscript deals with the new experimental data on the effects of various forms of polyamide microplastics to Daphnia magna survival and reproduction. Its content is clear. The manuscript is clearly written, the introduction yields sufficient background and is followed by research hypotheses, and the literature is adequately cited. English is clear and professional.

Experimental design

The experimental scheme is extremely simple. It includes the exposure of daphnids to virgin and UV-weathered polyamide microplastics at various concentrations of MP with or without the surfactant, in the close to natural conditions. Methods are described fully enough, just some details should be corrected (see general comments below).

Validity of the findings

The findings of the authors proved the importance of investigations on versatile effects of microplastics to aquatic biota. The observations were carried out properly, and the data are presented with adequate statistical analysis. Conclusions are reasonable and summarize the output of this study.

Additional comments

Table 1 can be placed in Supplemental materials because its data were not presented/discussed either in Results or Discussion parts of the ms.

Minor comments:
L. 28 (widely used plastic material). It can be omitted. Cf Line 19-20: "one of the commonly used MP worldwide."
L. 87 Change (Khosrovyan & Kahru (2021) to (Khosrovyan & Kahru, 2021).
L. 141. “In total, 6 experiments…” When looking into the Supplemental file, it is obvious that there were 6 treatments each performed within 2 (independent) experiments. Please correct.
L. 140–142. This paragraph should be joined with the above one.
L. 214 etc. Use the same spelling: “UV-weathering” throughout the text.
L. 283. Change (categories 1-… to (categories I-…
L. 335. Change ‘throughput’ to ‘throughout’.

---

## Round 0.2 · accepted · Accept

Thank you for revising the manuscript. It is now suitable for publication.

There are a couple of minor issues that I ask you to correct during the proof stage.

First, it is unclear if the question mark in the title is intended or was left there accidentally during revision of the previous title. In the response letter the authors state the new title without the question mark.

Second, there is a typographical error in Figure 3, in the title of y-axis. Please correct it during proof stage.